# Proof-Carrying Numbers (PCN): A Protocol for Trustworthy Numeric Answers from LLMs via Claim Verification

## Abstract

Large Language Models (LLMs) as stochastic systems may generate numbers that deviate from available data, a failure known as *numeric hallucination*. Existing safeguards—retrieval-augmented generation, citations, and uncertainty estimation—improve transparency but cannot guarantee fidelity: fabricated or misquoted values may still be displayed as if correct. We propose **Proof-Carrying Numbers (PCN)**, a presentation-layer protocol that enforces numeric fidelity through mechanical verification. Under PCN, numeric spans are emitted as *claim-bound tokens* tied to structured claims, and a verifier checks each token under a declared policy (e.g., exact equality, rounding, aliases, or tolerance with qualifiers). Crucially, PCN places verification in the *renderer*, not the model: only claim-checked numbers are marked as verified, and all others default to unverified. This separation prevents spoofing and guarantees fail-closed behavior. We formalize PCN and prove soundness, completeness under honest tokens, fail-closed behavior, and monotonicity under policy refinement. PCN is lightweight and model-agnostic, integrates seamlessly into existing applications, and can be extended with cryptographic commitments. By enforcing verification as a mandatory step before display, PCN establishes a simple contract for numerically sensitive settings: *trust is earned only by proof*, while the absence of a mark communicates uncertainty.

## 1 Introduction

Large Language Models (LLMs) are emerging as powerful interfaces for accessing knowledge in domains ranging from healthcare and finance to economics and international development. Their fluency makes them attractive to a wide range of users—from policymakers and researchers to clinicians, financial analysts, and the public—but their usefulness is constrained by their stochastic nature: they may generate **numeric hallucinations**.

Even when given correct input, LLMs may still produce plausible but incorrect values—sometimes citing the right dataset while presenting the wrong figure (Ji et al., 2023; Banerjee et al., 2024; Xu et al., 2025; Kalai et al., 2025). For example, Wu et al. (2025a) showed that when provided a perturbed drug dosage, an LLM sometimes "corrected" it to a different value. In another case, a model might state that the Philippines' GDP growth in 2024 was 6% when the official figure published by The World Bank (2025) was 5.7%. Small deviations like these can erode trust and cascade into flawed medical guidance, misinformed policy, or reputational risks for institutions.

Existing safeguards only partially address this problem. Retrieval-augmented generation (Lewis et al., 2020) grounds answers in source text, while citations and attribution frameworks (Wu et al., 2025a; Zhang et al., 2025) increase transparency. However, both remain probabilistic: users often assume a cited number is faithful even when it has been misquoted or fabricated (Wu et al., 2025b; Hakim et al., 2025). Similarly, uncertainty estimation (Manakul et al., 2023) and self-verification approaches can flag suspicious values but offer no binding guarantee.

**Proof-Carrying Numbers (PCN) Protocol in LLM Applications**

Figure 1: PCN-compliant architecture with LLM-initiated retrieval. The LLM first parses the query and *requests* claims from the retriever (dashed top lane); the retriever returns the claim set $C$ on a separate solid lane. The LLM emits claim-bound tokens (or bare numbers). The verifier checks tokens under policy $\Pi$, and the UI renders verified values with provenance marks; absence of a mark implies unverified by default. External structured data sources feed the retriever via parallel dashed feeders.

We argue that numeric hallucination is best understood as a **presentation-layer problem**. Even when authoritative claims are retrieved, LLMs are unreliable at reproducing values faithfully (Banerjee et al., 2024; Xu et al., 2025), and user interfaces lack systematic safeguards against drift or fabrication.

To address this gap, we propose **Proof-Carrying Numbers (PCN)**, a protocol that requires every displayed number to be bound to an authoritative claim and verified before presentation. Loosely inspired by proof-carrying code (Necula, 1997), PCN embeds verifiability directly into the interface: verified numbers carry explicit provenance, while unverifiable ones are blocked, flagged, or corrected.

Our contributions are threefold:

1. We reframe numeric hallucination as a *presentation-layer problem*, showing why existing safeguards (retrieval, citations, uncertainty estimation) cannot provide binding guarantees.

2. We design the **Proof-Carrying Numbers (PCN)** protocol, specifying a claim schema, token syntax, and verifier policies that enforce a fail-closed contract at display time.

3. We show how PCN wraps inherently fallible LLM outputs in a deterministic contract: numeric spans are either *Verified* against authoritative claims with provenance, remain *Bare* if unclaimed, or are *Flagged* when verification fails.

By embedding verification into the presentation pipeline, PCN bridges the gap between LLM fluency and the trustworthiness required in high-stakes numeric applications.

## 2  BACKGROUND AND RELATED WORK

Hallucination in large language models (LLMs)—the generation of fluent but incorrect content—poses serious challenges across domains (Ji et al., 2023; Banerjee et al., 2024; Xu et al., 2025). Of particular concern is *numeric hallucination*, where even small deviations (e.g., reporting 6.0% instead of 5.7%) can undermine high-stakes applications in policy, healthcare, and finance (Kim et al., 2025; Kang & Liu, 2023).

One stream of work grounds model outputs in retrieved content. Retrieval-augmented generation (Lewis et al., 2020) and citation frameworks (Wu et al., 2025a; Schreieder et al., 2025; Zhang et al., 2025) improve transparency by linking generated text to sources. However, fabricated values may still appear alongside credible references, creating the illusion of fidelity (Wu et al., 2025b; Hakim et al., 2025).

Another line of research focuses on post-hoc verification. Frameworks such as FEVER (Thorne et al., 2018), FEVEROUS (Aly et al., 2021), TabFact (Chen et al., 2020), and SciFact (Wadden et al., 2022) decompose outputs into claims and check them against evidence. Recent tools like AttributionBench (Li et al., 2024) and SourceCheckup (Wu et al., 2025a) extend this approach to LLMs. While useful for auditing, these methods are retrospective: they may flag erroneous outputs but cannot prevent unverified numbers from being displayed. Uncertainty-based methods (Manakul et al., 2023) similarly attempt to detect hallucinations using entropy (Farquhar et al., 2024), calibration (Manakul et al., 2023), or self-consistency (Kadavath et al., 2022), but confident models may still produce incorrect values with low uncertainty.

Structured decoding and symbolic grounding introduce additional constraints (Geng et al., 2023). Schema-constrained decoding enforces well-formed outputs, while symbolic methods such as SymGen (Hennigen et al., 2024) interleave generated text with explicit references to underlying data. These annotations reduce the burden of manual validation and make provenance more interpretable, but they stop short of guaranteeing fidelity: fabricated numbers can still appear structurally "valid" without being faithful.

At the data level, provenance frameworks like W3C Verifiable Credentials (W3C) use public key infrastructure to certify origin. While effective for data ingress, these assurances are lost once values are processed by an LLM, which may alter or fabricate outputs without detection.

In summary, prior work has improved transparency and auditing but cannot guarantee that the number ultimately displayed to the user is the one retrieved from an authoritative source. Proof-Carrying Numbers (PCN) addresses this gap by shifting from *annotation* and *detection* to *machine-enforced verification*. Under PCN, a number is marked as "verified" only if it is bound to an authoritative claim and passes deterministic checks at the presentation layer.

## 3 Problem Formalization

### 3.1 Context

Numeric hallucination is often framed as a retrieval problem, but many failures arise at the *presentation layer*: even when correct values are accessible, the number ultimately shown to the user may drift. PCN enforces a simple contract: a displayed numeric span is either Verified—because it can be mechanically matched to a structured claim under a declared policy $\Pi$—or it remains unverified (as *Bare* or *Flagged*). Verified marks thus provide positive guarantees, while their absence communicates uncertainty without suppressing content.

### 3.2 Setting

Consider a user query $q$ to an AI application that integrates structured data through a database, API, or Model Context Protocol (MCP) server (Anthropic). The application resolves $q$ into a finite claim set

$$C = \{c_1, c_2, \ldots, c_n\},$$

where each claim $c$ has the form

$$c = \langle \text{claim\_id}, \text{indicator}, \text{entity}, \text{time}, v^*, u, m \rangle.$$

Here $v^* \in \mathbb{R}$ is the reference value, $u$ the unit, and $m$ metadata (e.g., dataset version). Claims may optionally be cryptographically signed, though PCN does not assume a particular trust model.

### 3.3 Claim-Bound Tokens and Bare Numbers

An LLM generates an output sequence $y = (y_1, \ldots, y_T)$ that may contain numeric spans. PCN requires numeric values to be emitted as *claim-bound tokens* ($t$):

```
<claim id="CID" policy="P">VAL</claim>
```

where CID links to some $c \in C$, VAL is the displayed number, and P optionally specifies a verification policy. Such tokens bind surface text to structured claims.

By contrast, a *Bare number* is emitted without a claim tag. Since it cannot be linked to any $c \in C$, it is always treated as unverified. Bare numbers may still appear in text, but never carry a verification mark.

### 3.4 Verification Relation and Policies

To decide whether a token $t$ matches a claim $c$, PCN defines a verification relation $R(t, c; \Pi)$. Let $\hat{v}$ be the numeric payload of $t$ normalized into the claim's unit $u$. Verification modes supported by $\Pi$ include:

- **Exact match:** $R_{\text{exact}}(t, c) = 1 \iff \hat{v} = v^*$.
- **Rounded match:** for decimal precision $d$,
$$R^{(d)}_{\text{round}}(t, c) = 1 \iff \text{round}_d(\hat{v}) = \text{round}_d(v^*).$$
- **Alias equivalence:** for sanctioned scale/alias set $S$ (e.g., $\{10^3, \text{K}, \text{thousand}\}$),
$$R_{\text{alias}}(t, c) = 1 \iff \exists s \in S : \ \hat{v} \cdot s = v^*.$$
- **Tolerance with qualifiers:** for tolerance parameters $(\delta, \rho)$ and qualifier set $Q$ (e.g., $\{\text{"about", "approximately"}\}$),
$$R^{(\delta, \rho)}_{\text{tol}}(t, c) = 1 \iff \hat{v} \in [v^* - \max(\delta, \rho|v^*|), \ v^* + \max(\delta, \rho|v^*|)]$$
and $t$ includes a qualifier in $Q$.

A policy $\Pi$ specifies which relations are permitted. Formally,
$$R(t, c; \Pi) = 1 \iff \exists \text{ allowed mode in } \Pi \text{ such that it holds.}$$
If $t$ has no claim reference or $R(t, c; \Pi) = 0$, the number is treated as unverified.

### 3.5 Running Example

Suppose $C$ contains
$$c = \langle\text{"clm\_7ef6", GDP growth, PHL}, 2024, 5.7, \%, m\rangle.$$
If the LLM emits

```
<claim id="clm_7ef6" policy="round1">5.7</claim>
```

and $\Pi$ allows rounding to one decimal, verification succeeds since $\text{round}_1(5.7) = 5.7$. If the LLM emits

```
<claim id="clm_7ef6" policy="int">6</claim>
```

and $\Pi$ allows rounding to the nearest integer, verification again succeeds since $\text{round}_0(5.7) = 6$. By contrast, if the LLM emits 6.0 or even 5.7 without a claim tag, the number is *Bare* and displayed without a verification mark.

### 3.6 Problem Statement

Given a query $q$, a claim set $C$, and an output sequence $y$ with numeric spans $\{t_j\}$, the system must ensure

$\forall t_j \in y, \quad t_j$ is either verified against some $c \in C$ under policy $\Pi$, or surfaced as unverified.

The objective is to close the **presentation-layer verification gap**: every displayed number is either verifiably linked to a claim under $\Pi$ or left unverified by default.

## 4 PROPOSED APPROACH: PROOF-CARRYING NUMBERS

We introduce **Proof-Carrying Numbers (PCN)**, a protocol that enforces numeric fidelity by requiring that values shown to users carry verifiable links to structured claims. Building on the formalization in Section 3, PCN is not a decoding constraint but a *presentation-layer contract*: every displayed number is either mechanically verified against a claim or presented as unverified. This section describes PCN's architecture, verification policies, user contract, and possible extensions.

### 4.1 CONCEPTUAL OVERVIEW

PCN integrates verification into the rendering pipeline. Numeric spans generated by an LLM are annotated with claim references, checked against structured data, and rendered with explicit status indicators. This design closes the fidelity gap: LLMs can generate fluent text, but only numbers that pass verification are displayed with verified badges, while all others remain Bare or Flagged.

### 4.2 SYSTEM ARCHITECTURE

As illustrated in Figure 1, PCN consists of four lightweight components:

1. **Retriever:** resolves a query $q$ into a set of structured claims $C = \{c_1, \ldots, c_n\}$ from a data source such as a statistical database, financial API, medical record service, or MCP server.
2. **Generator:** produces an output sequence $y$ that may include claim-bound tokens (Section 3.3). Bare numbers may also appear, but they carry no proof.
3. **Verifier:** checks each token against the claim set under policy $\Pi$, succeeding if $R(t, c; \Pi) = 1$ and otherwise labeling the value as Bare or Flagged.
4. **User Interface:** renders Verified numbers with explicit provenance marks (e.g., a badge and hoverable metadata). Bare numbers appear without a mark, while Flagged values are shown with a warning indicator. The absence of a mark *by default* communicates that a number is not guaranteed.

This architecture is modular and lightweight, making PCN applicable to any system that integrates LLMs with structured data, regardless of retrieval protocol or model choice.

### 4.3 VERIFICATION POLICIES

Applications require different levels of strictness. PCN supports a range of policies, as defined in Section 3.4, including exact equality, rounding to specified decimal places, alias equivalence (e.g., "K" for thousands), and tolerance with qualifiers (e.g., "about," "roughly"). Policies encode an explicit trade-off: stricter rules provide higher trust but lower coverage, while permissive ones expand coverage at the cost of precision. This explicit policy layer distinguishes PCN from schema-based decoding, which constrains format but not correctness.

### 4.4 USER CONTRACT

PCN enforces a *fail-closed* contract. Users can rely on two guarantees:

1. Values marked as Verified have been mechanically checked against a claim under policy $\Pi$ and are displayed with provenance.
2. Values without such a mark are not verified, whether Bare (unclaimed) or Flagged (failed verification), and should be interpreted with caution.

This shifts the default assumption: current applications implicitly present all numbers as trustworthy, while PCN makes trust explicit and earned. This subtle change in user

experience is critical: it enables end-users—whether policymakers, clinicians, or financial analysts—to rely on verified numbers, distinguishing them from potential hallucinations.

### 4.5 Extensions: Cryptographic Proofs

PCN can be extended to settings requiring stronger provenance. Claims may embed cryptographic commitments such as Merkle proofs for large tables or PKI signatures for multi-provider trust chains. In such cases, verification not only checks numeric fidelity but also validates claim authenticity. These extensions strengthen tamper-evidence without altering the core contract: a number is verified only if it is mechanically tied to an authoritative claim.

## 5 Correctness Guarantees

We analyze the guarantees provided by PCN, given a generated sequence $y$, structured claim set $C$, acceptance policy $\Pi$, and verification relation $R(t, c; \Pi) \in \{0, 1\}$. The acceptance function is defined as:

$$A(y, C; \Pi) \mapsto \{(t_j, \text{label})\}_j$$

which labels each numeric span $t_j \in y$ as either Verified or Unverified.

### 5.1 Core Properties

**Theorem 5.1** (Soundness). *If $A(y, C; \Pi)$ labels $t$ as* Verified, *then there exists a claim $c \in C$ such that $R(t, c; \Pi) = 1$.*

*Proof sketch.* By construction, the verifier only assigns Verified if it finds such a claim. Hence no fabricated value can be marked as Verified. ∎

**Theorem 5.2** (Completeness under honest tokens). *If the generator emits a claim-bound token $t$ referencing some $c \in C$ and $R(t, c; \Pi) = 1$, then $A$ labels $t$ as* Verified.

*Proof sketch.* Determinism of the verifier ensures all policy-compliant tokens are accepted. ∎

**Theorem 5.3** (Fail-Closed). *Any span that (i) lacks a valid claim reference, (ii) references a non-existent claim, or (iii) fails verification under $\Pi$ is labeled* Unverified.

*Proof sketch.* The acceptance function defaults to Unverified unless an explicit match is found. ∎

**Lemma 5.4** (Monotonicity under policy refinement). *If $\Pi_1 \preceq \Pi_2$ (i.e., $\Pi_1$ is stricter), then:*

$$\{t : \text{Verified}_{\Pi_1}(t)\} \subseteq \{t : \text{Verified}_{\Pi_2}(t)\}.$$

*Proof sketch.* Tightening policies reduces coverage but never introduces false positives. ∎

**Implications.** Applications can expose multiple presets (e.g., *strict, rounded, approximate*) with predictable effects on the Verified set. Tightening a policy cannot introduce false positives; relaxing a policy cannot demote a previously verified token.

### 5.2 Robustness to Spoofing

**Theorem 5.5** (Renderer robustness). *If verification status is computed by the renderer rather than text tokens, then adversarial attempts to inject symbols (✓, "verified", HTML tags) into $y$ cannot cause $A$ to mislabel an* Unverified *token as* Verified.

*Proof sketch.* Verified status is derived solely from $R(t, c; \Pi)$. Spoofed tokens are ignored by the parser and remain Unverified. ∎

This property ensures that the mark itself is trustworthy and cannot be faked by prompt injection or adversarial text formatting.

## 5.3 EFFICIENCY

**Proposition 5.6** (Linear-time verification)**.** *Let $n \leq |y|$ be the number of numeric spans and $m = |C|$ the size of the claim set. If claims are indexed by identifier, then PCN verification runs in $O(n)$ time.*

*Proof sketch.* Each span lookup reduces to a hash-table access in $O(1)$. Policy checks are constant-time (rounding, alias lookup, tolerance check). ∎

This ensures PCN verification remains negligible compared to LLM generation latency.

## 5.4 CRYPTOGRAPHIC TAMPER-EVIDENCE (EXTENSION)

**Theorem 5.7** (Unforgeability of provenance)**.** *Assuming EUF-CMA security of the signature scheme and collision resistance of the hash, no adversary can cause A to label a tampered claim as* VERIFIED *except with negligible probability.*

*Proof sketch.* Verification requires a valid signature or Merkle proof. Forging this reduces to breaking standard cryptographic assumptions. ∎

## 5.5 SUMMARY

Together, these results show that PCN provides:

- **Correctness:** Verified numbers always correspond to claims (5.1–5.4).
- **Robustness:** Verification marks cannot be spoofed (5.5).
- **Efficiency:** Verification cost is negligible (5.6).
- **Security:** Tamper-evidence is cryptographically guaranteed (5.7).

Unlike heuristic methods, PCN requires no probabilistic confidence scoring. Fidelity follows deterministically from the protocol's construction.

# 6 DISCUSSION AND LIMITATIONS

Proof-Carrying Numbers (PCN) reframes numeric hallucination not as a question of whether the model "knows" the right value, but of what the user interface is permitted to display. By enforcing a fail-closed contract, PCN ensures that numbers marked as "verified" are mechanically tied to authoritative claims, while all others are visibly unverified. This shift moves trust from probabilistic model behavior to deterministic verification at the presentation layer.

## 6.1 SCOPE OF GUARANTEES

PCN's guarantees are intentionally modest but powerful. It does not claim that a model's reasoning is sound or that a dataset is "true." Instead, it guarantees that displayed numbers are either (i) verifiably consistent with a claim under a policy $\Pi$, or (ii) explicitly unverified. This closes one of the most dangerous loopholes in current AI systems: the undetected inclusion of fabricated numbers in fluent responses. In practice, this enables policymakers, clinicians, or analysts to treat verification marks as binding fidelity contracts, while interpreting unmarked numbers as provisional.

## 6.2 Policy Design and Usability

A defining feature of PCN is its *policy layer*, which governs how closely numeric spans must match reference claims to receive verification. While applications may define their own policies, data providers can also publish canonical rules (e.g., rounding conventions, tolerances) alongside claims. This reduces the risk of arbitrary or inconsistent application-level choices. For example:

- **Clinical dosage:** providers may require exact equality, reflecting zero tolerance for deviation.
- **Macroeconomic growth:** agencies may allow one-decimal rounding to match dissemination practices.
- **Journalistic communication:** datasets may permit approximate expressions ("about," "roughly") within a bounded tolerance.

Policies strengthen provenance but raise challenges of interoperability and accountability: strict rules may reduce coverage, while inconsistent ones across sources complicate multi-provider verification. The monotonicity property (Theorem 5.4) ensures such trade-offs remain predictable.

## 6.3 Risks and Threats

PCN's effectiveness depends on both human factors and adversarial resilience.

**Verification coverage gaps.** PCN only verifies numbers that the LLM tags with claim-bound tokens. If tagging recall is poor, many numeric spans will remain *Bare*—visible but without verification marks. This can create the perception that the system is unreliable, even though it reflects limitations in the LLM's compliance rather than the verifier itself. Improving prompting, fine-tuning, or constrained decoding is therefore essential to make PCN useful in practice.

**Policy misconfiguration.** Overly strict policies cause excessive verification failures, while permissive ones dilute guarantees. Since policies directly shape user trust, misconfiguration can either frustrate users or undermine fidelity. Clear presets (e.g., "strict," "tolerant") mitigate this risk.

**Overconfidence in scope.** PCN secures numeric fidelity only. Users may mistake badges for guarantees of holistic correctness, when reasoning and non-numeric facts remain outside its scope. Scope must be communicated explicitly.

**Institutional responsibility.** Verification ties numbers to specific providers, strengthening provenance but also shifting accountability. Custodians may hesitate to participate if they fear liability. Adoption will require governance frameworks that distribute responsibility across model providers, developers, and institutions.

**Adversarial considerations.** Common attack surfaces map directly onto PCN's guarantees: fabricated values cannot be marked (fail-closed), spoofed symbols do not confer verification (renderer robustness), and tampering with claim stores can be mitigated by cryptographic commitments. Additional operational safeguards (e.g., version pinning, audit logs) further reduce risk. Identifier abuse and privacy remain implementation-level considerations.

## 6.4 Integration and Overhead

From an engineering standpoint, PCN is lightweight. Verification runs in $O(n)$ time over numeric spans and adds negligible latency compared to model decoding. This makes it practical as a drop-in module for RAG pipelines, LLM-based chatbots, or statistical portals.

Early experiments suggest that prompting or light fine-tuning enables models to emit claim tags with reasonable recall, though further empirical work is needed.

### 6.5 Broader Implications

By decoupling fluency from fidelity, PCN reshapes incentives for trust in AI systems:

- **Developers** can build applications where trust derives from the verification pipeline rather than the model itself.

- **Institutions** (e.g., central banks, ministries of health, or the World Bank) can act as trust anchors by supplying authoritative claims.

- **Users** gain a clear signal: trust is earned only by proof, and the absence of a mark is itself informative.

This reframing does not solve factuality in general, but it addresses the class of errors with the highest downstream risk: misrepresented numbers. Even small numeric drifts can cascade into reputational or policy harms; PCN offers a minimal but enforceable safeguard.

### 6.6 Limitations and Future Work

PCN has clear boundaries: it cannot guarantee reasoning correctness, covers only structured claims, and depends on the availability of authoritative data sources. Its effectiveness also hinges on LLM cooperation in emitting claim-bound tokens. Future work should evaluate user behavior around verification marks, refine policy design trade-offs, extend PCN to derived values (e.g., ratios, aggregates) by verifying deterministic functions over atomic claims, and extend PCN to multi-provider cryptographic trust chains. Ultimately, PCN aims to make verification—not assumption—the default habit of numeric communication in AI.

## 7 Conclusion

We introduced *Proof-Carrying Numbers* (PCN), a protocol that makes numeric fidelity a presentation-layer property. PCN binds displayed numbers to structured claims and verifies each span under an explicit policy Π, yielding formal guarantees of soundness, completeness under honest tokens, and fail-closed behavior in which Bare or invalid spans never appear as Verified. Unlike retrieval, citation, or schema-only approaches, PCN treats verification as a first-class, mechanical step between model outputs and the user interface. The result is a simple contract: *trust is earned only by proof*, while the absence of a mark communicates uncertainty without suppressing content.

PCN is domain-agnostic: claims may originate from statistical databases, clinical systems, financial APIs, or other structured sources. Its modular architecture (retriever, generator, verifier, UI) integrates with existing applications, and optional cryptographic commitments (signatures, Merkle proofs) strengthen provenance without altering the core contract.

The protocol's scope is intentionally bounded. PCN guarantees correspondence to a chosen source, not ultimate truth, and it currently addresses atomic numeric spans rather than derived expressions or free-text facts. However, closing the verification gap at the presentation layer offers a practical step toward trustworthy numeric communication in LLM applications.

Future work includes developing SDKs and reference implementations across application stacks, designing adaptive policies such as tolerances with guardrails, and studying how users interact with verification marks and provenance cues. PCN could also be extended to cover derived values (e.g., ratios and aggregates) through deterministic functions over atomic claims, and to multi-provider deployments secured by cryptographic trust chains. Taken together, these directions position PCN as a minimal yet extensible blueprint for deploying LLMs in numerically sensitive settings with explicit, inspectable guarantees.

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

DISCLAIMER AND DISCLOSURE OF AI USE

Microsoft Co-Pilot and ChatGPT were employed to enhance the manuscript's readability and consistency. ChatGPT was also used to critique the previous versions of the manuscript for suggested improvements.

# A APPENDIX

## A.1 USER INTERFACE IMPLEMENTING PCN

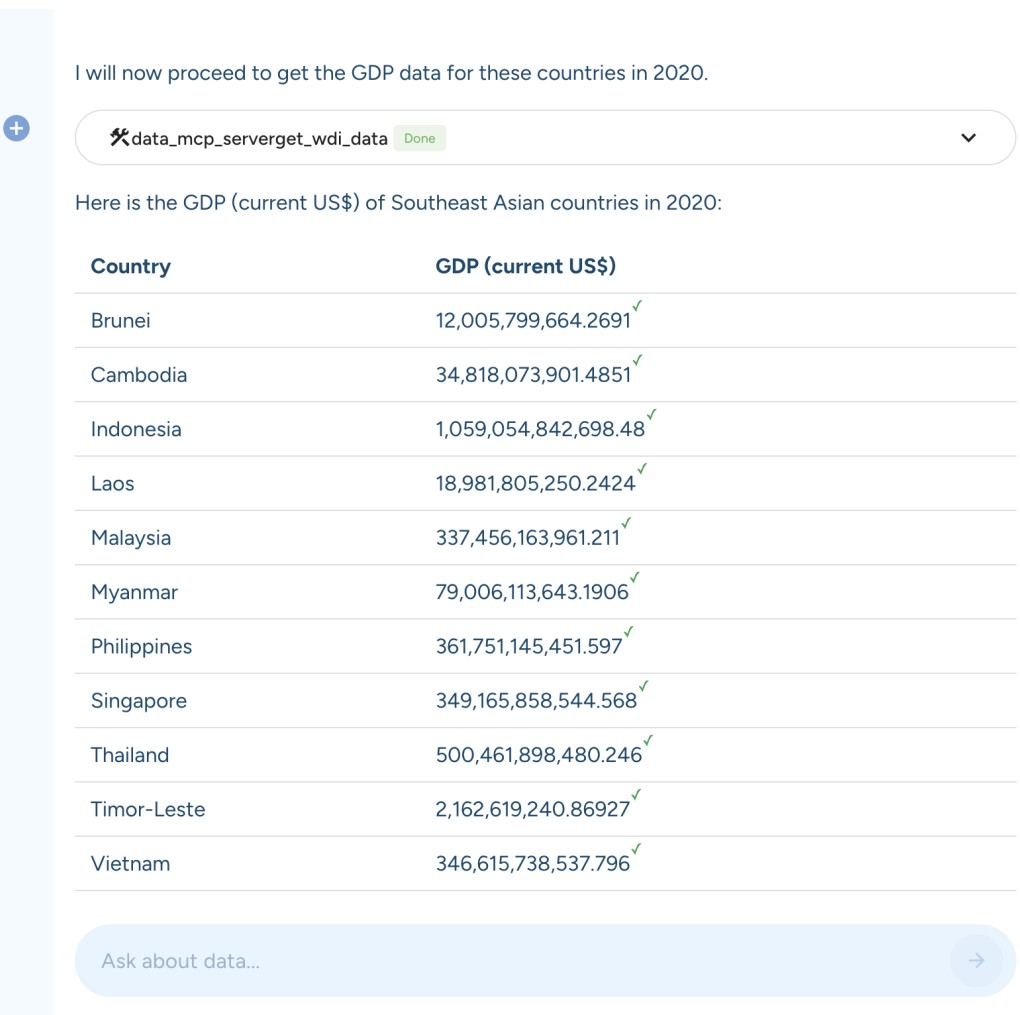

Figure 2: Implementation of the Proof-Carrying Numbers (PCN) protocol in an LLM chat application. The LLM retrieves authoritative claims via an MCP server; the verifier applies PCN policies; and the interface renders verified numbers with explicit badges.

## A.2 SAMPLE IMPLEMENTATION

Suppose the user asks a question to the LLM: "What is the gdp growth of the Philippines in 2024?"

The following shows how the Proof-Carrying Numbers (PCN) protocol works to provide an answer to the user.

First, the LLM queries the data retriever, in this case, an MCP server that gets data from the World Development Indicators. The MCP server implements the PCN protocol and returns this PCN payload with the claim to the LLM generator.

**PCN compliant claim from the retriever**

```json
{
  "data": [
    {
      "indicator_id": "NY.GDP.MKTP.KD.ZG",
      "indicator_name": "GDP growth (annual %)",
      "data": [
        {
          "country": "Philippines",
          "date": "2024",
          "value": 5.69201612823412,
          "claim_id": "0328"
        }
      ]
    }
  ],
  "note": {
    "NY.GDP.MKTP.KD.ZG": ""
  }
}
```

The LLM, given the payload, will generate a response that attempts to answer the user's question using the retriever's data. At the same time the LLM is instructed to conform to the PCN protocol in generating the response.

The instruction given to the LLM and its PCN-compliant response are shown below.

**Example prompt to encourage PCN use by the LLM**

```
[... other instructions]

When you provide any numerical data or values obtained from the tools, you must
  ↪  always enclose the numbers within a claim tag in the following format: `<claim
  ↪  id="claim_id">"value"</claim>`. For example, "The GDP of the Philippines in
  ↪  2020 is <claim id="5e1f">361,751,145,451.597</claim> USD". This is mandatory.
```

**PCN-compliant LLM response**

```
<p>The GDP growth rate of the Philippines in 2024 is projected to be <claim
  ↪  id="0328">5.69%</claim> (annual %).</p>
```

In parallel, when the receiver returns the PCN claim requested by the LLM, a claim aggregation process is applied to capture the claim, which can be used during verification.

The TypeScript function below shows how the PCN payload from the retriever is aggregated to create a map of claims that the verifier can use to verify the claims in the response generated by the LLM.

**Extraction of claims from the retriever component**

```typescript
export function getClaims (messages: ResponseMessage[]): Record<string,
  ↪  Record<string, any>> {
  const claims: Record<string, Record<string, any>> = {}

  for (const m of messages
    .filter(m => m.role === 'tool')) {
```

```
756
757        const parsed = m.toolContent?.output?.parsed
758        if (!parsed?.data) {
759          continue
760        }
761
762        for (const indicator of parsed.data) {
763          for (const d of indicator.data) {
764            if (d.claim_id) {
765              claims[d.claim_id] = {
766                country: d.country,
767                date: d.date,
768                value: d.value,
769                indicator_id: indicator.indicator_id,
770              }
771            }
772          }
773        }
774      }
775
776      return claims
777    }
```

Now that the LLM has generated a response, we execute the PCN-verifier module to assess
if claims have been made and then validate if any.

The TypeScript function below shows how the PCN protocol can be implemented to process
the response generated by an LLM. This example implements the exact policy variant.

### Processing of PCN claims in LLM content

```
const processPCNClaims = (content: string) => {
  return content.replace(
    /<claim id="([^"]+)">(.*?)<\/claim>/g,
    (match, claimId: string, innerText: string) => {
      // Remove any existing verification markers to make this idempotent
      const cleanInner = innerText.replace(
        /<sup class="(?:verified-mark|verify-pending)".*?<\/sup>/g,
        '',
      ).replace(
        /(.*?)<\/span>/g,
        '$1',
      )

      const claim = claims.value?.[claimId]
      if (!claim) {
        // No known claim for this id → mark as pending
        return `<claim id="${claimId}">${cleanInner}<sup class="verify-pending"
          ↪    title="Needs verification" role="img" aria-label="Needs
          ↪    verification">X</claim>`
      }

      // Normalize inner text & claim value (remove spaces/commas)
      const normalizedInner = String(cleanInner).replace(/[\s,]/g, '')
      const normalizedClaimValue = String(claim.value).replace(/[\s,]/g, '')

      if (normalizedInner === normalizedClaimValue) {
        // Verified
        return `<claim id="${claimId}">${cleanInner}<sup class="verified-mark"
          ↪    title="Verified data">OK</claim>`
      }

      // Mismatch → needs verification
```

```
        return `<claim id="${claimId}">${cleanInner}<sup class="verify-pending"
        ↪    title="${toTitleAttr(claim.country, claim.date, String(claim.value))}"
        ↪    role="img" aria-label="Needs verification">X</claim>`
    },
  )
}
```

The verifier modifies the content by injecting verification signals into the response. In this case, since the policy is exact, and the LLM opted to return a rounded-off version of the actual value received from the receiver, then the verifier returns a warning signal indicating the policy constraint was not met.

---

**Updated response by the verifier**

```
    <p>The GDP growth rate of the Philippines in 2024 is projected to be <claim
        ↪    id="0328">5.69%<sup class="verify-pending" title="Country: Philippines
Date: 2024
Value: 5.69201612823412" role="img" aria-label="Needs verification">X</claim>
        ↪    (annual %).</p>
```

---

I found relevant indicators for GDP growth rate for the Philippines. The most relevant indicator for GDP growth (annual %) is "GDP growth (annual %)" with the indicator code NY.GDP.MKTP.KD.ZG.

Next, I will retrieve the GDP growth rate data for the Philippines for the year 2024 if available.

🔧 data_mcp_serverget_wdi_data  `Done`                                        ︿

Arguments                                                    🔵 Parsed  📋

```json
{
  "indicator_ids": "NY.GDP.MKTP.KD.ZG",
  "country_codes_str": "PHL",
  "date": "2024",
  "per_page": 1
}
```

Output                                                       🔵 Parsed  📋

```json
{
  "data": [
    {
      "indicator_id": "NY.GDP.MKTP.KD.ZG",
      "indicator_name": "GDP growth (annual %)",
      "data": [
        {
          "country": "Philippines",
          "date": "2024",
          "value": 5.69201612823412,
          "claim_id": "0328"
        }
      ]
    }
  ],
  "note": {
    "NY.GDP.MKTP.KD.ZG": ""
  }
}
```

The GDP growth rate of the Philippines in 2024 is projected to be 5.69% ⚠ (annual %).

Figure 3: Illustration of the Proof-Carrying Numbers (PCN) protocol in an LLM chat application, where the response of the LLM (5.69%) didn't meet the exact policy set in the verification. A clear warning mark is added that hints to users to review the output.

