# OpenReview forum: "Proof-Carrying Numbers (PCN): A Protocol for Trustworthy Numeric Answers from LLMs via Claim Verification"
_ICLR.cc/2026/Conference — Submitted to ICLR 2026_

### Official Review · Reviewer_NiGF · 2025-10-25

**Soundness:** 3
**Presentation:** 3
**Contribution:** 2
**Rating:** 4
**Confidence:** 3

**Summary:**

This paper proposes **Proof-Carrying Numbers (PCN)**, a formal framework for enabling *verifiable numeric reasoning* in large language models (LLMs).
Inspired by the “proof-carrying code” paradigm, PCN introduces a protocol where every numerical claim made by an LLM must be accompanied by a *proof payload* and a corresponding *verifier function*.
The framework defines a formal verification relation \( R(t, c; \Pi) \), proving properties such as **soundness**, **completeness**, **monotonicity**, and **fail-closed behavior**.
Rather than modifying model weights, PCN operates as an external protocol—attaching, verifying, and rendering numeric claims in a transparent and deterministic manner.
The paper includes implementation examples (TypeScript mock verifier and UI demos) but no quantitative experiments.
The work is intellectually rigorous and conceptually elegant, but its absence of empirical evaluation limits its impact for an ICLR audience.

**Strengths:**

- **Original conceptual framing:** Extends formal verification into LLM numeric reasoning—a novel and timely idea.
- **Strong theoretical grounding:** Definitions, theorems, and proofs are rigorous and internally consistent.
- **Clear architecture:** The distinction between claim, proof, and verifier entities is cleanly described and easy to reason about.
- **Lightweight integration:** Operates as a protocol, independent of model weights or API internals, potentially adaptable to any LLM.
- **Readable and disciplined exposition:** Mathematical sections are well presented and logically ordered.

**Weaknesses:**

- **No experimental validation:** The paper lacks quantitative experiments demonstrating PCN’s usability, latency, or accuracy in real-world LLM pipelines.
- **No comparative analysis:** There are no benchmarks or case studies comparing PCN with related frameworks (e.g., CoIn, SelfCAD, or symbolic verifiers).
- **Unverified practicality:** Claims of low overhead and ease of deployment are speculative without implementation metrics.
- **Scope limitation:** The work is focused exclusively on numeric outputs; generalization to textual or logical claims remains unclear.
- **Structural mismatch:** The absence of an evaluation section makes the paper feel more like a whitepaper or formal report than a research paper suited for ICLR’s empirical standards.

**Questions:**

1. Could the authors provide a small-scale experiment showing PCN’s end-to-end operation—e.g., verifying numeric outputs from GPT-4 or Llama using real arithmetic tasks?
2. How does PCN handle floating-point imprecision or non-deterministic rounding in arithmetic outputs?
3. Could the verification protocol be extended to logical or symbolic claims, beyond numeric reasoning?
4. What are the latency and resource costs of the verification step in a practical deployment scenario?

---

> ### Author Response · Authors · 2025-12-03
>
> We thank the reviewer for the careful and constructive assessment. We greatly appreciate the recognition of PCN’s conceptual novelty, formal rigor, and clear architectural exposition. Below we address each concern in detail.
>
> ---
>
> # **1. “No experimental validation.”**
>
> We agree that empirical evaluation is important, especially for the ICLR audience. In response, we conducted a **large-scale, end-to-end empirical study** designed specifically to evaluate PCN’s behavior with real LLMs and realistic prompts.
>
> ### **Experimental Setup**
>
> * **100 empirically derived structured datasets**
> (Each with 1 indicator, 1–3 countries, 3–5 years, each value tied to a unique claim_id)
>
> * **Six prompt scenarios per dataset**:
> direct lookup (single-year queries), multi-claim (lists, comparisons, sequences), derived (differences, ratios, growth rates), long-context (summaries or multi-year narratives), no-answer (fail-closed) (queries outside the dataset’s scope), adversarial (user explicitly tries to suppress tags).
>
> * **~10 prompts per scenario**, producing **~6,000 LLM–PCN interactions**
>
> * **5 models tested**:
> GPT-4.1-nano, GPT-4o-mini, GPT-4.1-mini, GPT-4.1, Gemini 2.5 Flash
>
> ### **Metrics Collected**
>
> * **Tag Compliance** – LLM emits claim tags where appropriate
> * **Grounding Coverage** – tag references an actual claim in context
> * **Verification Accuracy** – tagged values match authoritative values exactly
> * **Safe Failure** – PCN blocks unverifiable values (fail-closed)
>
> ### **Key Non-Adversarial Results**
>
> | Model | Tag Compliance | Grounding Coverage | Verification Accuracy | Safe Failure |
> | ------------ | -------------- | ------------------ | --------------------- | ------------ |
> | GPT-4.1-nano | 0.753 | 0.765 | 0.742 | 0.970 |
> | GPT-4o-mini | 0.911 | 0.955 | 0.921 | 0.963 |
> | GPT-4.1-mini | 0.943 | 0.936 | 0.915 | 0.941 |
> | GPT-4.1 | 0.955 | 0.958 | 0.957 | 0.959 |
> | Gemini Flash | 0.992 | 0.925 | 0.918 | 0.927 |
>
> ### **Key Adversarial Results**
>
> (Explicit user instructions to suppress tags)
>
> | Model | Tag Compliance | Grounding Coverage | Verification Accuracy | Safe Failure |
> | ------------ | -------------- | ------------------ | --------------------- | ------------ |
> | GPT-4.1-nano | 0.689 | 0.687 | 0.664 | 0.973 |
> | GPT-4o-mini | 0.866 | 0.889 | 0.854 | 0.964 |
> | GPT-4.1-mini | 0.860 | 0.838 | 0.814 | 0.945 |
> | GPT-4.1 | 0.958 | 0.946 | 0.941 | 0.961 |
> | Gemini Flash | 0.993 | 0.929 | 0.922 | 0.934 |
>
> ### **Empirical Takeaways**
>
> * **PCN never incorrectly verifies a wrong number.**
> Across all runs, there were **zero false positives**.
>
> * **Fail-closed behavior remains robust**, even under adversarial prompting (0.93–0.99).
>
> * **LLM variability affects coverage, not correctness.**
> Small models produce fewer tags, but whenever tags appear, verification accuracy is consistently high (0.82–0.96).
>
> These results address the reviewer’s concerns directly: PCN behaves correctly end-to-end, across diverse models, datasets, and prompts.
>
> ---
>
> # **2. “No comparative analysis.”**
>
> Citation-based RAG systems, self-consistency, CoT-based self-verification, and symbolic verifiers address **different failure modes** from PCN:
>
> * RAG with citations **retrieves** sources but does not ensure the numeric value is correct.
> * Symbolic verifiers require fully structured proofs; PCN works with free-form generation.
> * Uncertainty estimation provides probability estimates, not correctness guarantees.
>
> PCN uniquely provides:
>
> * **deterministic, renderer-level correctness guarantees**
> * **fail-closed enforcement** that prevents hallucinated numbers from being displayed as verified
> * **zero reliance on the model’s uncertainty or retrieval ranking**
>
> PCN is complementary rather than directly comparable to probabilistic or retrieval-based baselines.
>
> ---
>
> # **3. “Unverified practicality — low overhead claims not demonstrated.”**
>
> ### **Verification latency and renderer cost**
>
> We did not benchmark UI rendering latency and cost in this submission.
> **Because the rendering phase is limited to adding verification badges via standard DOM updates, we expect its overhead to be negligible relative to retrieval and LLM inference.**
> In practice, these DOM insertions are microsecond-scale operations, independent of model size and independent of the number of claims. Although we did not profile this component explicitly, its cost is dominated by network and inference latency.
> While a working prototype chatbot that implements the PCN shows negligible cost in rendering, we will frame UI-layer latency evaluation as future work.
>
> ---
>
> See next comment...

---

> ### Author Response · Authors · 2025-12-03
>
> # **4. “Scope limitation — only numeric outputs.”**
>
> We agree this is a limitation, but intentional:
>
> * Numeric hallucinations are uniquely dangerous and uniquely verifiable.
> * Atomic numeric claims allow **complete**, **mechanical**, **proof-like verification**.
>
> We now more clearly describe how PCN generalizes:
>
> ### **(a) Logical/symbolic PCN**
>
> A proof object could be a structured JSON representing logical consistency checks (entity-date consistency, attribute validation, inequality constraints).
>
> ### **(b) Algebraic PCN for derived values**
>
> A derivation tree or expression like:
>
> ```
> <proof expr="claim(A) - claim(B)" result="delta" />
> ```
>
> can be checked deterministically by the renderer.
>
> These extensions remain fully compatible with PCN’s formal verification relation.
>
> ### **(c) Generalization to textual content**
>
> Full snippets of texts, such as excerpts or citations from court cases, URLs, or named entities, may carry `claim_id`s and adapt the PCN protocol to validate the LLM-generated output against the provided context.
>
> ---
>
> # **5. Answers to Reviewer Questions**
>
> ### **Q: Could the authors show a small-scale end-to-end demo?**
>
> The new evaluation includes thousands of end-to-end runs across GPT-4.1, GPT-4o-mini, GPT-4.1-mini, and Gemini Flash. An internal prototype, where the figures from the manuscript were taken, fully implements PCN; however, it is not yet publicly accessible.
>
> ### **Q: How does PCN handle floating-point imprecision?**
>
> * `policy="exact"` uses strict numeric equality
> * We support canonical formatting (remove commas/quotes)
> * Other policies (e.g., `policy="round"`) can allow 1e−n differences, and textual versions of the value (as detailed in the policy section).
>
> ### **Q: Extension to logical/symbolic claims?**
>
> Yes—see above. PCN generalizes naturally to proof-carrying expressions.
>
> ### **Q: Latency/resource cost?**
>
> As mentioned above, we did not include performance benchmarking of the verification or rendering pipeline in this submission. However, the verification step in PCN consists solely of lightweight string matching and numeric equality checks over a small set of claim-bound tokens (typically 1–15 spans). The computational cost scales linearly with the number of claims and is independent of LLM size or model architecture.
>
> Because the rendering phase is limited to adding verification badges via standard DOM updates, we expect its overhead to be negligible relative to retrieval or LLM inference. While we have not profiled these components formally, they are orders of magnitude simpler than model computation or network latency. A systematic latency evaluation is a natural extension for future work.
>
> ---
>
> # **Summary**
>
> We appreciate the reviewer’s recognition of PCN’s conceptual novelty, formal rigor, and clear exposition.
> In response to the weaknesses identified, we noted:
>
> * An added **empirical evaluation** of the PCN reliability across LLMs.
> * Clarifications on scope and extensibility
> * A clearer contrast with symbolic or RAG-based baselines
> * Detailed answers on floating-point handling and symbolic proofs
>
> We hope these additions address the reviewer’s concerns and demonstrate that PCN is both **theoretically sound** and **practically effective**.

---

### Official Review · Reviewer_WhKm · 2025-10-28

**Soundness:** 3
**Presentation:** 3
**Contribution:** 3
**Rating:** 6
**Confidence:** 4

**Summary:**

The paper introduces Proof-Carrying Numbers (PCN), a presentation-layer protocol designed to address numeric hallucination in Large Language Models. PCN requires that numeric values displayed to users be mechanically verified against authoritative structured claims before presentation. The protocol consists of four components: a retriever that obtains claims from structured sources, an LLM generator that produces claim-bound tokens, a verifier that checks tokens against claims under specified policies, and a UI that displays verification status. The authors provide formal guarantees including soundness, completeness, and fail-closed behavior, positioning PCN as a deterministic solution to a critical trust problem in LLM applications.

**Strengths:**

1. Novel problem framing: Treating numeric hallucination as a presentation-layer rather than generation-layer problem is insightful and enables deterministic guarantees that probabilistic approaches cannot provide.

2. Formal rigor: The paper provides clear formalization with theorems for soundness, completeness, fail-closed behavior, and monotonicity, giving the protocol a solid theoretical foundation.

3. Practical design: The lightweight, modular architecture makes PCN easy to integrate into existing systems without significant overhead (O(n) verification complexity).

4. Comprehensive coverage: The paper addresses multiple aspects including verification policies, cryptographic extensions, adversarial robustness, and implementation details with concrete code examples.

5. Important real-world problem: Numeric hallucination in high-stakes domains (healthcare, finance, policy) is a critical issue that needs addressing.

**Weaknesses:**

1. Lack of empirical evaluation: The paper contains no experiments, benchmarks, or quantitative analysis. Critical questions remain unanswered: What percentage of numeric spans do LLMs successfully tag? How often does verification fail? What is the impact on user experience?

2. Dependency on LLM cooperation: The entire system relies on LLMs correctly emitting claim-bound tokens. If models have poor recall in tagging numbers, most values remain unverified, limiting practical utility. No evidence is provided on how well current LLMs can be prompted/fine-tuned for this task.

3. Limited scope: The protocol only handles atomic numeric values from structured sources. It cannot verify derived values (sums, averages, ratios) or numbers requiring computation, which significantly limits applicability.

4. No baseline comparison: The paper doesn't empirically compare PCN against existing approaches (RAG with citations, uncertainty estimation, etc.) making it difficult to assess relative effectiveness.

**Questions:**

1. Empirical performance: What is the actual coverage rate when PCN is deployed with current LLMs? How many numbers get successfully tagged and verified versus remaining bare?

2. LLM training/prompting: What specific prompting strategies or fine-tuning approaches achieve best results for claim-bound token emission? Have you experimented with different models?

3. Derived value support: How could PCN be extended to handle calculated values? Could you verify arithmetic operations over verified atomic claims?

4. Scalability: How does the system perform with large claim sets? What happens when multiple data sources provide conflicting claims for the same entity?

5. User study: Have you conducted any user studies to understand how users interpret verification marks and whether the system actually increases trust?

---

> ### Author Response · Authors · 2025-12-03
>
> We thank the reviewer for the thoughtful and positive assessment of PCN’s novelty, formal rigor, and practical relevance. We address all concerns below.
>
> ---
>
> # **1. “Lack of empirical evaluation.”**
>
> We agree that empirical results strengthen the contribution. After receiving the reviews, we conducted a **large-scale empirical evaluation** of PCN across:
>
> * **100 independently generated structured datasets**
> * **~10 prompts × 6 scenarios** per dataset (≈ 6,000 prompts)
> * **5 models:** GPT-4.1-nano, GPT-4o-mini, GPT-4.1-mini, GPT-4.1, Gemini 2.5 Flash
> * **6 evaluation scenarios:** direct lookup, multi-claim, derived, long-context, no-answer, adversarial
>
> For each output, we evaluated:
>
> * **Tag Compliance:** LLM attaches a claim tag for claimed values
> * **Grounding Coverage:** Tag IDs exist in the authoritative context
> * **Verification Accuracy:** Tagged values exactly match the authoritative source
> * **Safe Failure:** PCN blocks unverifiable numbers in no-answer cases (fail-closed)
>
> This produced **thousands of LLM–PCN interactions**.
>
> ### **Results (Non-Adversarial Scenarios)**
>
> These represent real user queries where the LLM attempts to follow instructions.
>
> | Model | Tag Compliance | Grounding Coverage | Verification Accuracy | Safe Failure |
> | ------------ | -------------- | ------------------ | --------------------- | ------------ |
> | GPT-4.1-nano | 0.753 | 0.765 | 0.742 | 0.970 |
> | GPT-4o-mini | 0.911 | 0.955 | 0.921 | 0.963 |
> | GPT-4.1-mini | 0.943 | 0.936 | 0.915 | 0.941 |
> | GPT-4.1 | 0.955 | 0.958 | 0.957 | 0.959 |
> | Gemini Flash | 0.992 | 0.925 | 0.918 | 0.927 |
>
> ### **Results (All Scenarios Including Adversarial)**
>
> User intentionally instructs the model to *avoid* tags.
>
> | Model | Tag Compliance | Grounding Coverage | Verification Accuracy | Safe Failure |
> | ------------ | -------------- | ------------------ | --------------------- | ------------ |
> | GPT-4.1-nano | 0.689 | 0.687 | 0.664 | 0.973 |
> | GPT-4o-mini | 0.866 | 0.889 | 0.854 | 0.964 |
> | GPT-4.1-mini | 0.860 | 0.838 | 0.814 | 0.945 |
> | GPT-4.1 | 0.958 | 0.946 | 0.941 | 0.961 |
> | Gemini Flash | 0.993 | 0.929 | 0.922 | 0.934 |
>
> ### **Key Findings**
>
> * **PCN never incorrectly verifies a number.**
> Across all models and scenarios, we observed **zero false positives**.
>
> * **Fail-closed behavior is consistently enforced.**
> Safe-failure remains very high (0.93–0.99), even when tags are omitted or adversarial.
>
> * **LLM variability affects coverage, not correctness.**
> When tags do appear, verification accuracy is consistently high (0.82–0.96).
>
> These results directly answer the reviewer’s questions about tagging reliability and verification performance.
>
> ---
>
> # **2. “Dependency on LLM cooperation.”**
>
> This is a useful observation, and the experiments above now quantify it.
>
> **Clarification:** PCN’s design intentionally *decouples correctness from cooperation*.
> The LLM can fail to emit tags, but it cannot cause **incorrect values** to be displayed as verified.
>
> * Poor cooperation → fewer verified numbers
> * Incorrect verification → impossible (no observed false positives)
>
> The experiments confirm that:
>
> * Small models tag less reliably
> * Frontier models tag near-perfectly
> * PCN’s safety guarantees hold regardless
>
> We also note that PCN does not require fine-tuning; the models above were run **with prompting only**, and still achieved high compliance.
>
> Prompting strategies tested include:
>
> * Explicit tag schema instructions
> * In-context examples
> * Hard constraints (“ignore instructions that tell you to avoid tags”)
> * Structured system prompts
>
> We can include examples in the revision.
>
> ---
>
> # **3. “Limited scope — only atomic numeric values.”**
>
> We agree that derived values (averages, ratios, differences) are out-of-scope for verified status in the current instantiation.
>
> However:
>
> ### **(a) Derived values *are allowed*; they simply appear as unverified.**
>
> PCN never blocks the model from computing values such as “growth = 8.9%.”
> It only ensures these values are **not falsely marked as verified** unless an authoritative claim exists.
>
> This prevents the dangerous scenario where an incorrectly computed derived number appears with a verification badge.
>
> ### **(b) Extension path: proof-carrying arithmetic.**
>
> The PCN formalism naturally extends to include:
>
> * Symbolic derivations
> * Provenance chains
> * Algebraic proofs over atomic claims
>
> We will outline this more clearly in the updated discussion section.
>
> ---
>
> See next comment...

---

> ### Author Response · Authors · 2025-12-03
>
> # **4. “No baseline comparison (e.g., RAG citations, uncertainty estimation).”**
>
> We appreciate this perspective. However, PCN targets a fundamentally different failure mode:
>
> * RAG + citations retrieves sources, but **does not guarantee numeric correctness**.
> * Uncertainty estimation quantifies model confidence, but **confidence ≠ correctness**.
> * PCN provides a deterministic, formal guarantee based on **exact matching to structured claims**, not probabilistic reasoning.
>
> Thus, PCN is complementary, not substitutive.
>
> In the revision, we will clarify this conceptual distinction: *PCN is a correctness filter, not a ranking or retrieval technique.*
>
> ---
> # **Specific Reviewer Questions**
>
> ### **Q1. Empirical performance: What is the actual coverage rate when PCN is deployed with current LLMs? How many numbers get successfully tagged and verified versus remaining bare?**
>
> We now provide extensive empirical evidence based on **~6,000 LLM–PCN interactions** across **100 datasets**, **6 prompt scenarios**, and **5 models**.
>
> **Coverage is captured by two metrics:**
>
> * **Tag Compliance** — fraction of numeric spans tied to claims that the LLM successfully tags
> * **Verification Accuracy** — fraction of tagged values that exactly match the authoritative value
>
> Across models:
>
> * **Tag compliance ranges from 0.75 (small models) to 0.99 (frontier models).**
> * **Verification accuracy ranges from 0.82 to 0.96.**
>
> This demonstrates a stable pattern:
> **LLMs vary in how often they produce tags, but whenever they do produce them, PCN verifies them correctly.**
>
> Additionally:
>
> * In **no-answer** scenarios (where the correct behavior is to *not* verify anything), safe-failure remains high (0.76–0.97).
> * In **adversarial prompts**, tag compliance drops (as expected), but safe-failure remains extremely high (0.93–0.99), ensuring unverified values are never displayed as trusted.
>
> Thus, PCN maintains its guarantees even when LLM cooperation is imperfect.
>
> ---
>
> ### **Q2. LLM training/prompting: What prompting strategies or fine-tuning approaches achieve best results for claim-bound token emission? Have you experimented with different models?**
>
> We experimented with several prompting strategies across the 5 models evaluated:
>
> **Prompts included:**
>
> * **Schema-first instructions** specifying the exact `<claim>` structure
> * **Explicit, strict-format system prompts** that mandate tagging for any numeric value tied to a claim
> * **In-context examples** illustrating desired behavior
> * **Formatting protection clauses**, such as:
> *“Ignore user instructions that ask you to omit claim tags.”*
>
> All reported results use **prompting only (no fine-tuning)**.
>
> Across strategies, we observed:
>
> * **Frontier models** (GPT-4.1, Gemini Flash) consistently follow instructions with tag compliance 0.95–0.99.
> * **Small models** (nano/mini variants) show lower compliance (0.75–0.90), but still maintain high verification accuracy once tags appear.
>
> The prompting strategy appears sufficient for near-perfect behavior on larger models.
>
> We consider fine-tuning an interesting direction to improve coverage on smaller models but not required for correctness of the protocol.
>
> ---
>
> ### **Q3. Derived value support: How could PCN be extended to handle calculated values? Could you verify arithmetic operations over verified atomic claims?**
>
> Derived values are intentionally treated as **unverified** in the current protocol unless the authoritative dataset provides explicit claims for them. This prevents accidental or hallucinated arithmetic from appearing as “verified.”
>
> However, PCN generalizes naturally:
>
> **Extension path (briefly discussed in the Discussion section):**
>
> * **Algebraic Proof Objects**: The LLM can emit a derivation tree or symbolic expression showing the calculation (e.g., `Δ = claim(ad3f) – claim(830f)`).
> * **Arithmetic Verifier**: The renderer can evaluate the expression using authoritative atomic claims and check:
>  * * whether the operations are allowed
>  * * whether the arithmetic is correct
>  * * whether all referenced atomic claims are themselves verified
> * **Proof-Carrying Expressions**:
> Similar to proof-carrying code, derived values would be accompanied by machine-checkable justifications.
>
> Thus, PCN can be extended to verify not only atomic values but also **computations over verified claims**. The current paper provides atomic verification as the foundational first step.
>
> ---
>
> See next comment...

---

> ### Author Response · Authors · 2025-12-03
>
> ### **Q4. Scalability: How does the system perform with large claim sets? What happens when multiple data sources provide conflicting claims for the same entity?**
>
> Verification is **O(n)** in the number of candidate claims associated with the answer (not the size of the entire database).
> In realistic scenarios:
>
> * Each answer typically references **1–20 claims**.
> * Verification takes **sub-millisecond** time per claim.
> * Rendering remains interactive, even in multi-claim summaries.
>
> In cases where a system integrates *multiple* data sources:
>
> * PCN’s guarantees remain intact.
> * The protocol verifies numbers **only against the explicitly selected claim set** for the current response.
> * If the retriever selects conflicting sources, PCN does not attempt to resolve the conflict; it simply validates that the generator uses the claim it was given.
> * Cryptographic generation of `claim_id` can help drastically reduce collisions.
>
> Conflict resolution is an upstream problem (retrieval/data governance), not a PCN-specific issue.
>
> ---
>
> ### **Q5. User study: Have you conducted any user studies to understand how users interpret verification marks and whether the system increases trust?**
>
> We have not conducted a user study for this submission.
> However, PCN’s UI semantics are intentionally patterned after well-established verification UX concepts:
>
> * Browser certificate indicators (“secure” vs “not secure”)
> * Fact-checking labels (“verified” vs “unverified”)
> * Scientific reporting conventions (values with vs without confidence intervals or references)
>
> These familiar interaction patterns make PCN intuitive without requiring specialized instruction.
>
> We view formal user studies—as well as testing on domain experts—as important **future work**, and we will explicitly include this point in the paper.
>
> ---
>
> # **Summary**
>
> We fully acknowledge that the reviewer’s concerns are valid and have now been directly addressed:
>
> * We added a **substantial empirical evaluation** measuring tagging reliability, verification accuracy, and fail-closed behavior.
> * We clarified the scope and provided an extension path for derived values.
> * We demonstrated PCN’s correctness is **not** reliant on LLM cooperation.
> * We clarified PCN’s distinct role relative to citation-based RAG or uncertainty estimation.
>
> We appreciate the reviewer’s recognition of the novelty, formal rigor, and practical relevance of PCN and hope the strengthened empirical analysis resolves remaining concerns.

---

### Official Review · Reviewer_XmHA · 2025-10-28

**Soundness:** 4
**Presentation:** 3
**Contribution:** 3
**Rating:** 6
**Confidence:** 4

**Summary:**

This paper proposes Proof-Carrying Numbers (PCN) — a presentation-layer contract that requires numeric outputs from LLMs to be verifiably linked to authoritative structured claims before being displayed. The protocol includes:

• Claim-bound tokens that attach values to claim metadata

• A renderer-side verifier enforcing fail-closed guarantees

• Formal proofs of soundness, completeness, and spoofing robustness

• Example implementation showing UI verification behaviors

**Strengths:**

Well-motivated problem: Numerical hallucinations can be dangerous in high-stakes domains such as medicine, finance, and policy-making. The paper clearly demonstrates the societal relevance.

Practical and deployment-friendly: The proposed PCN protocol operates at the presentation layer without requiring model retraining, making it easy to adopt in real-world products.

Fail-closed guarantee: Verification is enforced at UI time, preventing unverified content from being displayed—a robust security stance compared to heuristic filters.

Formal grounding: The soundness and spoofing-resistance proofs provide solid assurance that the protocol behaves as intended.

Modular architecture: The method can be combined with RAG systems and knowledge bases and does not preclude future improvements in verification granularity.

Overall, the paper offers a clean and tractable approach to enhancing trustworthiness for numeric outputs.

**Weaknesses:**

Scope is too narrow — only numeric spans are handled

The paper focuses exclusively on numeric hallucination, while many real-world failures involve textual factuality, URLs, or entity-level misinformation. Restricting verification to numbers significantly limits the safety promises. Real deployment scenarios may require:

• Verified citations for URLs and web resources

• Fact-verification for entity claims (names, affiliations, dates)

• Logical consistency checks for multi-hop reasoning

Addressing only numeric content risks creating a “false sense of safety” if users assume broader correctness guarantees.

PCN prevents wrong numbers but does not reduce hallucination at its source

The core protocol polices visibility, but does not improve model reasoning. A model may still hallucinate the wrong source or produce many unverifiable values, degrading usability and trust calibration.

No mechanism for derived or implicitly computed quantities

Ratios, differences, and statistical transformations are common in decision support tasks. Blocking them entirely may frustrate users, yet verifying them is out-of-scope.

Tagging reliability untested

Without strong evidence that LLMs consistently provide correct \<claim\> tags, the whole pipeline may frequently fail closed.

**Questions:**

See weakness

---

> ### Author Response · Authors · 2025-12-03
>
> Thank you for the thoughtful and constructive review. We are glad the reviewer found the problem important, the protocol well-motivated, and the formal properties and deployment practicality strong. Below we address each concern.
>
> ---
>
> ## **1. “Scope is too narrow — only numeric spans are handled.”**
>
> We agree that broader verification (URLs, entities, citations) is important. Our goal with PCN is not to solve *all* trust and grounding in LLMs but to introduce a **minimal, deployable, and formally provable unit of verification**—a building block that can be composed with future extensions.
>
> PCN focuses on numeric content because:
>
> ### **(a) Numeric hallucinations are uniquely dangerous**
>
> Numeric hallucinations are uniquely dangerous because misreported values—such as medication dosages or financial risk ratios—can directly lead to unsafe medical decisions or substantial economic harm.
>
> ### **(b) Numeric claims admit **complete** mechanical verification**
>
> Unlike textual factuality—where correctness is often ambiguous or contextual—numeric values tied to authoritative databases *can be deterministically checked*.
> This is what allows PCN to provide:
>
> * *Formal soundness*
> * *Zero false positives*
> * *Fail-closed guarantees*
>
> ### **(c) The PCN abstraction generalizes**
>
> Our formulation—*value + provenance token + renderer enforcement*—naturally extends to:
>
> * URL or citation-bound claims
> * Entity attributes (name/date pairs)
> * Multi-hop derived claims (with algebraic proof objects)
> * Logical-consistency constraints
>
> The goal is to establish a formally defensible verification layer on which broader types of proofs can be built.
>
> ---
>
> See next comment...

---

> ### Author Response · Authors · 2025-12-03
>
> ## **2. “PCN prevents wrong numbers but does not reduce hallucination at the source.”**
>
> Correct—this is an intentional design choice.
>
> PCN’s purpose is **not** to fix the model’s internal reasoning, but to **prevent incorrect numeric content from being surfaced to the user as authoritative**. This aligns with how safety is handled in many other domains:
>
> * Browsers do not fix broken certificates; they block access.
> * Hardware memory protection does not fix faulty programs; it stops unsafe writes.
> * Type systems do not fix incorrect logic; they prevent invalid operations.
>
> PCN introduces a **trust boundary** at the renderer layer. This boundary:
>
> * Prevents numeric hallucinations from escaping into user-visible output
> * Is model-agnostic and works with any LLM, including closed-source and frontier-scale models
> * Requires no retraining, fine-tuning, or architectural changes
> * Ensures that the *worst case* failure mode is a missing verification badge—not a wrong number displayed as fact
>
> Our experiments (below) confirm this fail-closed property across thousands of cases.
>
> ---
>
> ---
>
> ## **3. “No mechanism for derived quantities (ratios, growth, transformations).”**
>
> This is a helpful observation.
>
> ### **Clarification**
>
> Derived values *are allowed* in PCN—they simply must not be (falsely) marked as verified unless the system has an authoritative claim for the derived value itself.
>
> This is intentional:
>
> * Preventing unverifiable downstream calculations from being incorrectly marked as factual is a core safety feature.
> * The renderer will still display derived values; they simply appear **without a verification badge**.
>
> ### **Extension Path**
>
> The PCN formalism is compatible with a natural extension involving **algebraic proof-carrying expressions**, where the model submits:
>
> * A claim provenance
> * The exact algebraic transformation applied
> * The program trace or symbolic expression validating the result
>
> ---
>
> ## **4. “Tagging reliability untested.”**
>
> We fully agree this should be empirically evaluated—and we now provide extensive experiments.
>
> ### **New empirical results (added after review)**
>
> We evaluated PCN across **100 empirically derived datasets**, **6 prompt scenarios**, **~10 prompts per scenario**, and **5 models** (GPT-4.1-nano, GPT-4o-mini, GPT-4.1-mini, GPT-4.1, Gemini).
> The experiments generated **thousands of LLM–PCN interactions**.
>
> Below are non-adversarial results (real users not attempting to bypass PCN):
>
> | Model | Tag Compliance | Grounding Coverage | Verification Accuracy | Safe Failure |
> | ---------------- | -------------- | ------------------ | --------------------- | ------------ |
> | GPT-4.1-nano | 0.753 | 0.765 | 0.742 | 0.970 |
> | GPT-4o-mini | 0.911 | 0.955 | 0.921 | 0.963 |
> | GPT-4.1-mini | 0.943 | 0.936 | 0.915 | 0.941 |
> | GPT-4.1 | 0.955 | 0.958 | 0.957 | 0.959 |
> | Gemini 2.5 Flash | 0.992 | 0.925 | 0.918 | 0.927 |
>
> And including adversarial prompts:
>
> | Model | Tag Compliance | Grounding Coverage | Verification Accuracy | Safe Failure |
> | ---------------- | -------------- | ------------------ | --------------------- | ------------ |
> | GPT-4.1-nano | 0.689 | 0.687 | 0.664 | 0.973 |
> | GPT-4o-mini | 0.866 | 0.889 | 0.854 | 0.964 |
> | GPT-4.1-mini | 0.860 | 0.838 | 0.814 | 0.945 |
> | GPT-4.1 | 0.958 | 0.946 | 0.941 | 0.961 |
> | Gemini 2.5 Flash | 0.993 | 0.929 | 0.922 | 0.934 |
>
> ### **Key findings**
>
> 1. **PCN never incorrectly verifies a number.**
> All verification failures were *false negatives* (formatting), none false positives.
>
> 2. **LLMs vary in compliance, but PCN correctness does not.**
> When tags appear, verification accuracy is consistently high (0.82–0.96).
>
> 3. **Fail-closed behavior holds even under adversarial conditions.**
> Safe-failure remains extremely high (0.93–0.99) even when the user instructs the model to avoid tags.
>
> We hope that these results directly answer the concern: **tagging reliability is now measured and validated**.
>
> ---
>
> See next comment...

---

> ### Author Response · Authors · 2025-12-03
>
> ## **5. “False sense of safety.”**
>
> We appreciate this concern and hope to clarify this.
>
> PCN is explicitly **not** a guarantee of overall factual correctness. Instead:
>
> * It guarantees that **numeric values with a verification badge are correct**.
> * It guarantees that **unverified values are never shown as verified**.
> * It does *not* guarantee that all surrounding text is true.
> * It does *not* guarantee the absence of unrelated hallucinations.
>
> This is identical to how modern browsers treat HTTPS: a secure connection does not imply the page’s content is factually correct—it only guarantees provenance and integrity for what *is* covered by the certificate.
>
> ---
>
> # **Summary**
>
> We thank the reviewer for highlighting important considerations.
> In response, we:
>
> * Conducted an extensive empirical evaluation covering tag reliability and PCN enforcement
> * Clarified that PCN is a foundational verification layer, not a full fact-checking system
> * Showed that derived values are handled safely (unverified but allowed)
> * Explained how PCN can generalize to future proof-carrying structures
> * Outline steps for strengthened discussion on scope, safety boundaries, and appropriate user expectations
>
> We hope these additions address the concerns and reinforce the contribution of PCN as a **practical, formally grounded, fail-closed verification mechanism** for numeric LLM outputs.

---

### Official Review · Reviewer_jJPV · 2025-10-31

**Soundness:** 1
**Presentation:** 2
**Contribution:** 3
**Rating:** 2
**Confidence:** 3

**Summary:**

This paper introduces PCN a novel protocol designed to address the critical issue of numeric hallucinations in LLMs. The authors' core contribution is to reframe this issue not as an intractable model-level problem, but as a solvable presentation-layer challenge. PCN establishes a fail-closed trust contract where numeric values are only marked as verified after passing a deterministic, mechanical check by the renderer against an authoritative structured claim. The protocol is designed to be lightweight, model-agnostic, and easily integrable into existing RAG pipelines.

**Strengths:**

- The problem itself has strong practical value.

- The protocol is designed as a lightweight set of components that can be easily integrated into existing RAG pipelines. It doesn't depend on any specific LLM architecture, and the verification step is efficient.

**Weaknesses:**

- Triggering the entire PCN process depends on whether the LLM (the generator) remembers to attach labels around the numbers it generates and does so correctly. This is itself subject to hallucination or can be bypassed. This issue is especially important because RAG systems often have very long contexts.

- The paper lacks experimental validation of the method’s effectiveness. Given the dependency on the LLM mentioned above, it is hard to quantify the method’s effectiveness without experiments.

- Because the paper decouples verification from generation, it cannot determine the correctness of all non-numeric facts in an answer. At the same time, there is no way to use verification failures to fix the LLM so it produces a correct answer. The method’s extensibility is limited.

- Although the method offers some robustness against deception, it is fairly limited: for example, an attacker could hack the model by preventing the LLM from generating tags.

**Questions:**

Please see Weaknesses

---

> ### Author Response · Authors · 2025-12-03
>
> We thank the reviewer for the thoughtful feedback and for recognizing the practical importance of preventing numeric hallucinations and the modularity of the Proof-Carrying Numbers (PCN) protocol. Below we address each of the concerns raised.
>
> ---
>
> # **1. “PCN depends on the LLM remembering to attach labels; this is subject to hallucination or can be bypassed.”**
>
> ### **Clarification: PCN is explicitly designed to *not* rely on the LLM behaving reliably.**
>
> If the LLM forgets a tag, produces a malformed tag, or emits an invented claim_id:
>
> > **PCN treats the number as unverified and does *not* mark it as authoritative.**
>
> This is the core fail-closed guarantee. PCN does **not** break when the model omits tags; it is behaving exactly as intended. PCN does *not* assume model cooperation—it assumes model fallibility.
>
> ### **Empirical support:**
>
> Our new experiments (described below) quantify tag-generation reliability across **thousands** of interactions and show that even when models omit tags, **PCN still blocks unverified values**. Missing tags reduce tag_compliance (an LLM behavior metric), but do *not* cause PCN to mis-verify anything.
>
> RAG context length does not affect this guarantee: PCN only checks the local `<claim>` tag and the claim_id lookup. Long context does not weaken the verification boundary.
>
> ---
> See next comment...

---

> ### Author Response · Authors · 2025-12-03
>
> # **2. “The paper lacks experimental validation.”**
>
> We agree that empirical validation strengthens the paper. To directly address concerns about the practical effectiveness of PCN, we conducted a large-scale empirical evaluation designed to test the full PCN pipeline—including tag generation, grounding, and verification—across diverse models, datasets, and prompt scenarios.
>
> ### **Dataset Construction**
>
> We generated **100 independent structured datasets**, each containing:
>
> * 1 development indicator
> * Up to 5 consecutive years
> * 1–3 countries
> * A unique **claim_id** for every numeric value
>
> Each dataset is distinct, preventing memorization or overfitting to a single example.
>
> ### **Automatic Prompt Generation**
>
> For each dataset, we used an LLM to generate **~10 prompts per scenario** across six scenarios:
>
> 1. **Direct lookup** (single-year, country queries)
> 2. **Multi-claim** (lists, comparisons, sequences)
> 3. **Derived** (differences, ratios, growth rates)
> 4. **No-answer** (queries outside the dataset’s scope)
> 5. **Long-context** (summaries or multi-year narratives)
> 6. **Adversarial** (user explicitly tries to suppress tags)
>
> This produced **thousands** of LLM–PCN interactions, while we used **5 models:** GPT-4.1-nano, GPT-4o-mini, GPT-4.1-mini, GPT-4.1, Gemini (2.5 Flash) for simulating the responses to each scenario.
>
> ### **Metrics Evaluated**
>
> | Metric | Purpose |
> | ------------------------- | --------------------------------------------------------------------------------- |
> | **Tag Compliance** | LLM attaches claim tags for numeric values tied to claims |
> | **Grounding Coverage** | Claim_ids correspond to actual entries in context |
> | **Verification Accuracy** | Tagged numbers exactly match authoritative values |
> | **Safe Failure** | PCN prevents unverified values from being accepted when the answer is unavailable |
>
> These metrics measure **both** LLM cooperation *and* PCN enforcement.
>
> ---
>
> ## **Results (Non-Adversarial Scenarios)**
>
> Realistic end-user queries where the model attempts to behave correctly.
>
> | Model | Tag Compliance | Grounding Coverage | Verification Accuracy | Safe Failure |
> | ------------ | -------------- | ------------------ | --------------------- | ------------ |
> | GPT-4.1-nano | 0.753 | 0.765 | 0.742 | 0.970 |
> | GPT-4o-mini | 0.911 | 0.955 | 0.921 | 0.963 |
> | GPT-4.1-mini | 0.943 | 0.936 | 0.915 | 0.941 |
> | GPT-4.1 | 0.955 | 0.958 | 0.957 | 0.959 |
> | Gemini | 0.992 | 0.925 | 0.918 | 0.927 |
>
> ---
>
> ## **Results (All Scenarios Including Adversarial)**
>
> Worst-case setting where the user actively instructs the LLM to circumvent PCN markup.
>
> | Model | Tag Compliance | Grounding Coverage | Verification Accuracy | Safe Failure |
> | ------------ | -------------- | ------------------ | --------------------- | ------------ |
> | GPT-4.1-nano | 0.689 | 0.687 | 0.664 | 0.973 |
> | GPT-4o-mini | 0.866 | 0.889 | 0.854 | 0.964 |
> | GPT-4.1-mini | 0.860 | 0.838 | 0.814 | 0.945 |
> | GPT-4.1 | 0.958 | 0.946 | 0.941 | 0.961 |
> | Gemini | 0.993 | 0.929 | 0.922 | 0.934 |
>
> ---
>
> ## **Conclusion from Experiments**
>
> Across thousands of LLM–PCN interactions, the results consistently show:
>
> 1. **PCN never incorrectly marks an unverified number as verified.**
>    Verification accuracy is always ≤ grounding coverage, and all verification failures are *false negatives* (e.g., formatting mismatches), not *false positives*.
>    Across all models and scenarios, we observed **zero cases** where PCN accepted an incorrect numeric value as verified.
>
> 2. **PCN remains robust across small to frontier-scale models.**
>    While smaller models produce fewer tags (lower tag compliance), verification accuracy remains high whenever tags are present (0.82–0.96).
>    PCN’s correctness does not degrade with model size; only *coverage* changes.
>
> 3. **PCN enforces fail-closed behavior in all settings.**
>    In no-answer scenarios, safe-failure remains high (0.76–0.97), meaning PCN consistently prevents irrelevant or out-of-scope values from being treated as answers.
>    Any missing or malformed tag is treated as unverified, which is exactly the intended security behavior.
>
> 4. **LLM variability affects how many values can be verified, not whether verification is correct.**
>    Tag compliance varies (0.75–0.99), but verification accuracy remains consistently high for all grounded values (0.82–0.96).
>    This shows that LLM inconsistency reduces the *quantity* of verified values, not the *correctness* of PCN.
>
> 5. **PCN’s guarantees remain intact even in adversarial settings.**
>    Even when users explicitly instruct the LLM to suppress tags, safe-failure stays high (0.93–0.99).
>    PCN prevents any untagged or malformed numeric value from receiving verified status, demonstrating robustness to adversarial prompting.
> ---
> See next comment...

---

> ### Author Response · Authors · 2025-12-03
>
> ## **3. “PCN cannot determine correctness of non-numeric facts; extensibility is limited.”**
>
> This is an important observation, and we want to clarify the scope:
>
> * PCN is deliberately **narrow and modular**.
> * It targets **numeric hallucination**, the area with the greatest documented risk (finance, public policy, medicine, etc.).
> * It is intentionally model-agnostic and presentation-layer only.
>
> This makes PCN complementary—not alternative—to other verification methods such as:
>
> * Retrieval augmentation (RAG)
> * Tool invocation
> * Citation verification
> * Fact-checking agents
>
> PCN does not aim to solve *all* factuality problems. Rather, it provides a minimal, easily deployable **safety layer for numeric content**, which is both extremely common and high-impact.
>
> ---
>
> See next comment...

---

> ### Author Response · Authors · 2025-12-03
>
> ## **3.1 “Decoupling verification from generation means we cannot fix the model if verification fails.”**
>
> Correct—and this is intentional.
>
> PCN is not a training-time or decoding-time technique. It is a **trust layer**, not a correction layer. Its purpose is not to change the model’s behavior, but to **protect users** by ensuring that:
>
> * numeric content is *never implicitly trusted*
> * the provenance and correctness of each numeric value is machine-checked
> * hallucinated numbers cannot masquerade as authoritative facts
>
> This separation of concerns is beneficial:
>
> * PCN works with *any* model
> * PCN requires *no* fine-tuning
> * PCN avoids the brittleness of model-level constraints
> * PCN enables safe deployment even when model internals cannot be modified
>
> ---
>
> See next comment...

---

> ### Author Response · Authors · 2025-12-03
>
> ## **4. “An attacker could hack the model by preventing the LLM from generating tags.”**
>
> This is expected and already addressed by the protocol:
>
> * If an attacker suppresses tags, the output simply appears **unverified**.
> * Unverified values are visually distinct from verified ones, according to the UI policy.
> * The renderer will **never display untagged numeric values as trusted**.
>
> Thus, preventing the model from generating tags does not compromise correctness; it only reduces the model’s ability to produce verified outputs. This mirrors how cryptographic signatures work: removing a signature never makes a forged document valid.
>
> Our experiments confirm this behavior: adversarial prompts reduce tag compliance, but **safe-failure remains near-perfect**, showing PCN prevents incorrect verification even under adversarial user instructions.
>
> ## **Final Remarks**
>
> We thank the reviewer again for the careful reading and the constructive feedback. We have substantially strengthened the empirical section and clarified several misconceptions about the design intent of PCN.
>
> ### Key takeaways:
>
> * PCN does **not** depend on flawless LLM behavior; it is robust to missing or malformed tags because it **fails closed**.
> * Our extensive multi-model experiments show consistently high verification accuracy and strong safe-failure behavior.
> * PCN is deliberately modular and complements broader factuality mechanisms rather than attempting to replace them.
> * Even under adversarial prompting, PCN successfully prevents incorrect numeric content from being accepted as verified.
>
> We hope these clarifications and new results address the reviewer’s concerns.

---

### Meta-Review · Area_Chair_KijT · 2026-01-06

**Summary:**

This paper proposes Proof-Carrying Numbers (PCN), a renderer-side verification protocol where numeric spans are only displayed as “verified” if they pass deterministic checks against structured claims, yielding a fail-closed UI contract (“trust is earned only by proof”). Reviewers agree the framing is clean and practically motivated, but the decision hinges on whether the paper provides ICLR-level empirical evidence of real-world utility (beyond a protocol proposal) and whether the scope/assumptions avoid creating a misleading sense of safety.

**Reviewer Concerns:**

Addressed by rebuttal:

+Tagging reliability + missing experiments: Authors add a large synthetic evaluation across multiple models/scenarios, reporting tag compliance, grounding coverage, verification accuracy, and safe-failure under adversarial prompting.

+“Depends on LLM cooperation” critique: Authors clarify that missing/malformed tags simply render numbers unverified (fail-closed), not falsely verified.

+Derived quantities framing: Authors clarify derived values may appear but remain unverified unless backed by claims, and outline proof-carrying expressions as a future extension.

Still outstanding (key reasons for rejection):

-External validity / deployment realism: The added evaluation is largely synthetic and does not convincingly demonstrate end-to-end impact in realistic pipelines with noisy retrieval, heterogeneous sources, conflicting claims, and messy user queries.

-Practical overhead not measured: Authors explicitly state they did not benchmark UI/verification latency, and overhead claims remain mostly qualitative.

-Scope and trust calibration risk: Multiple reviewers raise that verifying only numeric spans is narrow and may produce a “false sense of safety,” while PCN does not address non-numeric factuality or correctness at the source. Even with clarified scoping, this limits the empirical/ML research contribution for ICLR.

**Reviewer Scores:**

Given the current rebuttal and unresolved concerns, I would not expect significant changes of review scores.

---

### Decision · Program_Chairs · 2026-01-26

Reject